# Indoor Floor Heel Mark Removal Using Spark Discharges and Pressurized Airflow

Yoshihiro Sakamoto [1,2,*], Takayoshi Tsutsumi [2], Hiromasa Tanaka [2], Kenji Ishikawa [2], Hiroshi Hashizume [2] and Masaru Hori [2]

1    Center for Core Technology Development, Panasonic Co., Ltd., Kusatu 525-8555, Japan
2    Center for Low-Temperature Plasma Sciences, Nagoya University, Furo-cho, Chikusa-ku, Nagoya 464-8601, Japan
*    Correspondence: sakamoto.yo@jp.panasonic.com

**Abstract:** Heel marks (HMs), which are the black stains made by shoe soles on indoor floors, can be difficult to remove. However, this study shows how spark discharges combined with pressurized airflow in 60 s discharge treatments can remove such HMs. We further show that maximizing the HM removal rates depended on the electrode gap distance because of changes in the spark discharge parameters. In our experiments, the electrical voltage waveforms are shown with voltage spikes, called spark discharges, and the spike numbers were counted in 0.6-ms time units. It was found that the number of spark discharges increases when the electrode gap distance was widened from 5 mm to 10 mm and the pressurized airflow was added, and the HM removal rates increased 11.5%, the HM removal rates could be maximized. Taken together, the results show that spark discharges combined with pressurized air can remove HMs from indoor floors without no visual damage. This paper is a preliminary report showing that HMs can be removed by plasma.

**Keywords:** heel marks (HM); spark discharge; polyvinyl chloride (PVC)





## 1. Introduction

The indoor floors of airports, hotel lobbies, shopping malls, and many other buildings are often soiled by difficult-to-remove black stains, called heel marks (HMs), which are caused by the impacting shoe soles of people walking. Such HMs consist of mechanically bonded polymer organic residues such as polyurethane (-O=CN-R-NC=O-) and nylon (-NH-R-C=O-). The indoor floors most affected by such stains are often lined with sample flooring tiles that consist of a clear surface coat and a base layer containing a plasticizer and polyvinyl chloride (PVC) [1].

Once HMs have bonded to such floor tiles, they can normally only be removed by scrubbing them off with abrasive chemical cleaners or mechanical floor polishers. However, since these processes can result in floor damage and the production of chemical waste, the development of a new method that can rapidly remove stubborn HM stains is required.

To date, electrical discharge plasma technologies have been used in a variety of fields, such as food decontamination [2], plasma electrolysis, air purification, etching [3–10], semiconductor processes [11–14], and polymer film removal [15–18]. In our previous studies, non-equilibrium atmospheric pressure plasma was applied to glass surface cleaning [19,20], copper reduction [21], as well as silicon dioxide (SiO2) [22], and polymer etching [23]. There are many reports of atmospheric low-temperature plasma, but most of them use rare gases such as He and Ar, and few use Air. Air plasma can be used to sterilize plants and foods [24,25] and to modify the surface of polyamide [26], no research has been reported to remove "harsh stains" such as HM. The cleaners using atmospheric low-temperature plasma [27] used oxygen gas in vacuum to clean fingerprints, photoresist and pump oil in 2 min, which are better removed with O radicals than with Air, but air was reported to be ineffective. Since in this course of this study, which targets the development of a

cleaning method and an airport cleaner that does not require the use of detergents and causes no visual damage to the floor surface, it is essential that the gas used for the plasma is air. we first attempted to use plasma jets [28,29] and piezo plasma [30,31] to provide cleaning action. However, since those processes were unable to remove HMs effectively, we turned our focus to the application of spark discharges [32]. First, the distance between the discharge electrodes and the airflow supply effect was investigated. Next, the number of spark discharges in the current-voltage waveform was counted and correlated with the observed HM removal rates. Our new finding is that harsh stain of HM can be removed in 60 s by the air plasma technology at atmospheric pressure and this cleaning efficiency can be controlled by the number of spark discharges.

This is the first study to demonstrate that motivate the originality of this study is that stubborn stains like HM can be removed with Air-plasma. We are considering and have obtained brief results for removal of magic ink, sebum and cooking vegetable oils, as examples for difficult removal without detergents in the past. Hence, the general purpose of this technique is applicable not only HM on the floor but also on strongly modified organic stains.

## 2. Experimental Section

### 2.1. Sample Preparation

The floor samples used in this study were IS 899 PVC floor tiles (Sangetsu Corp., Sapporo, Japan) consisting of a 300 μm clear surface layer and a 2.5 mm base layer that included a plasticizer.

HMs was manufactured by grinding the heel of the shoe at a pressure of 12 kgf/cm$^2$ against the floor sample used in the experiment. The actual measurement confirms that the HMs thickness was 5–15 μm. A photograph showing typical HMs bonded to a floor sample is shown in Figure 1a, while a schematic cross-section of the floor sample is shown in Figure 1b.

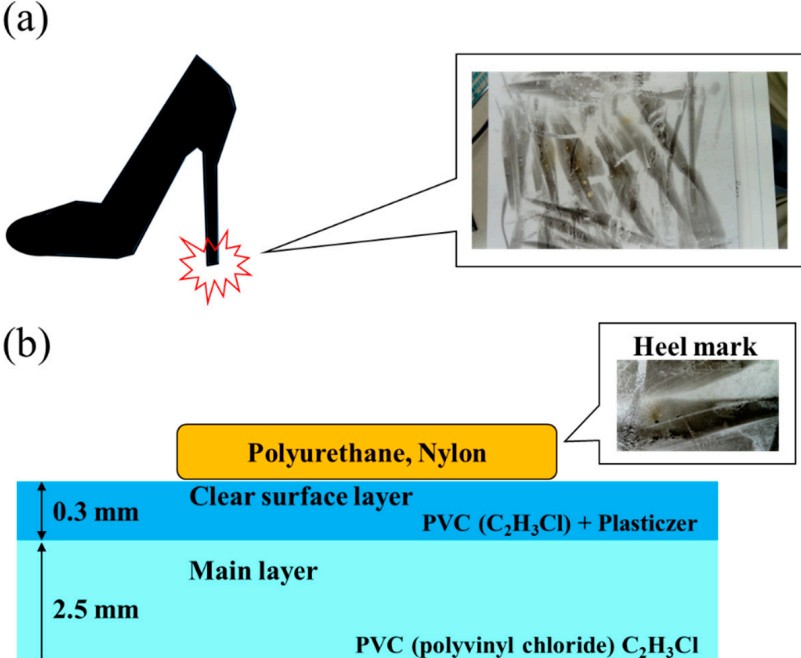

**Figure 1. Typical stain examples.** (**a**) HMs are produced by scratching or scrubbing a shoe sole on the floor tiles. (**b**) The cross-sectional structure of a sample floor tile showing the HM, clear surface layer, and main layer.

### 2.2. Experimental Setup

Figure 2 shows a schematic overview of an atmospheric pressure plasma treatment system used in our experiments. Two electrodes were placed at distances of 5 and 10 mm from each other. (Hereafter, this parameter is referred to as the gap). Both electrodes were fabricated from AWG 22 Sn- and Bi-coated 0.65 mm-diameter copper wire. The AWG 20 electrode wire on the power supply side was connected via a copper cable, while the electrode wire on the GND side was connected via a 1.25-mm$^2$ VSF cable (KHD Electronics Co., Ltd., Hirakata, Japan).

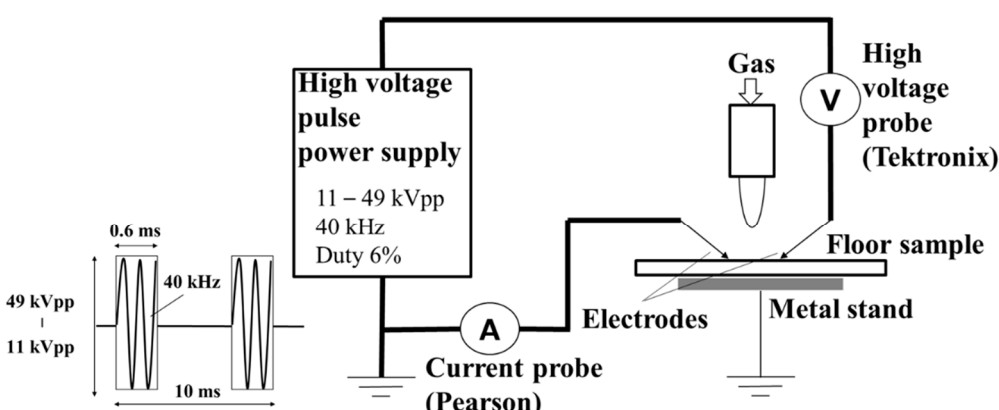

**Figure 2. Schematic overview of experimental system.** A high-voltage pulsed power supply is connected to the electrode. The high-voltage and current probes were used to detect the voltage and current waveforms, respectively. A sample floor tile was placed on an electrically grounded plate. The pressurized airflow was provided from tubing placed above the sample, and the flow rate was controlled by a flow meter.

Both electrodes were positioned approximately parallel to the floor sample surface, where pressurized dry air was emitted from a 2 mm diameter nozzle placed 35 mm above and between them. A typical flow rate of 20 standard-liter-per-minute (SLM) was used in our experiments. A metal stand was placed on a metal plate and connected to ground (GND). The three conditions (A), (B), and (C) used in our experiments are listed in Table 1.

**Table 1.** Experimental conditions.

| Condition | Electrode Gap (mm) | Airflow (20 SLM) |
|:---:|:---:|:---:|
| A | 5 | no |
| B | 10 | no |
| C | 10 | yes |

The electrodes were supplied with high-voltage electricity from a PVM500-4000 pulse power supply (Advanced Circuitry International, Duluth, GA, USA) to generate plasma. A sinusoidal frequency of 40 kHz was modulated by a square pulse waveform with a 6% duty of the turn-on period of 0.6 ms. The voltage levels of 11.0, 20.0, 28.0, 33.2, 38.0, 40.0, 42.6, 44.2, 47.4, and 49.0 kVpp were applied in our experiments. Unless otherwise noted, the treatment time was fixed at 60 s.

### 2.3. Analysis

The floor surface was observed using an SU 8230 scanning electron microscope (SEM) (Hitachi High-Tech, Tokyo, Japan). The SEM images of Figures 3d–f and 4 were observed at electron acceleration energy of 1 kV and an electron current of 5 µA, at a working distance of 8 mm. Magnifications are 2000 for Figures 3d–f and 4m–p, 500 for Figure 4i–l, and 100 for Figure 4e–h. The SU8230 was equipped with an X-Max50 energy dispersive X-ray

detector (Oxford Instruments, Oxford, UK). The elemental components analysis of the floor sample was performed using the SU8230 with X-Max50(SEM-EDX) which was electron acceleration voltage of 15 kV, at a working distance of 15 mm, magnification of 200. The floor cross-section was observed using a JSM-7900F SEM (Jeol Ltd., Tokyo, Japan). The SEM images of Figure 5b,c were observed at electron acceleration energy of 2 kV and electron current of 65 μA, at a working distance of 6 mm, magnification of 10,000×. Note that Figure 3f shows the surface condition of Figure 5c. The photographs shown in Figure 3a–c, Figure 4a–d, and Figure 5a were taken with an iPhone SE 15.5 (Apple Inc., Cupertino, CA, USA).

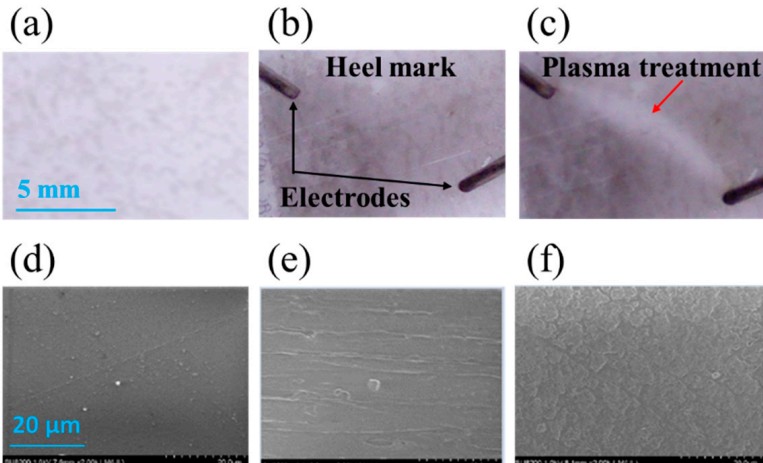

**Figure 3. Photographs and SEM results on floor surface with HMs.** (**a**) Unmarked (control) floor surface, (**b**) HMs bonded to the floor surface with two electrodes placed 10 mm apart, and (**c**) the floor tile during the plasma treatment. The SEM images were taken at magnification of 2000. (**d**) Unmarked (control) floor surface, (**e**) HMs bonded to the floor surface, and (**f**) the floor surface after HM removal of plasma treatment for 60 s.

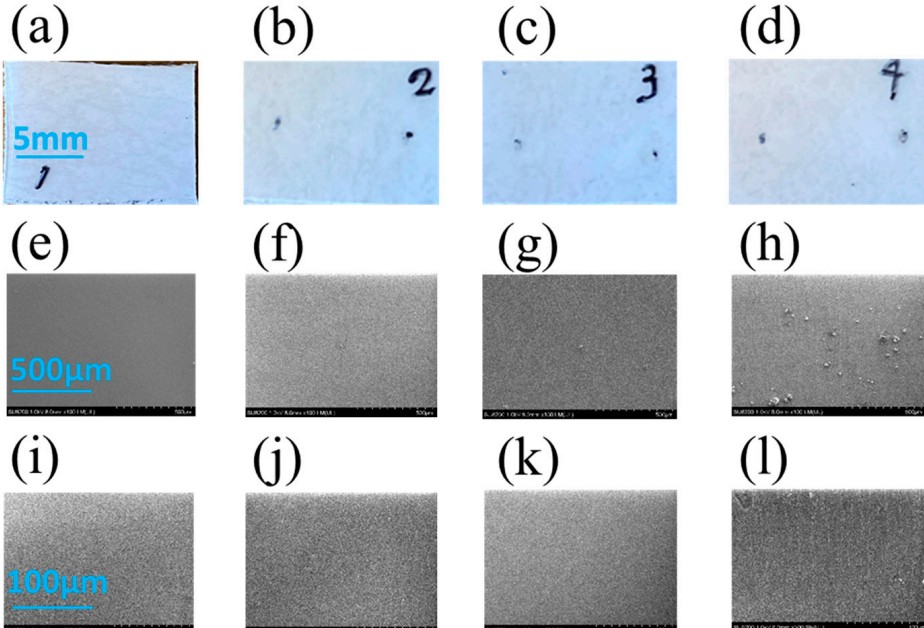

**Figure 4.** *Cont.*

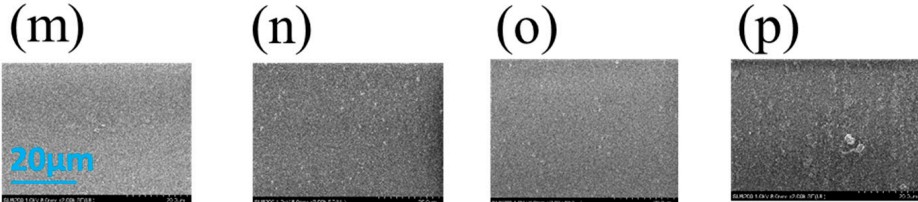

**Figure 4. Photographs (a–d) and SEM images (e–p) on floor surface without HMs.** (**a–d**) two black marks on the floor indicate the locations where the two electrodes were placed 10 mm apart, (**a,e,i,m**) Control floor surface, (**b,f,j,n**) The plasma-treated floor surface for 60 s. (**c,g,k,o**) The floor surface via plasma treatment for 180 s, and (**d,h,l,p**) The floor surface via plasma treatment for 300 s. (**e–h**) SEM images of (**a–d**) were taken at magnification of 100 for (**i–l**), 500 for (**a–d**), and 2000 for (**m–p**).

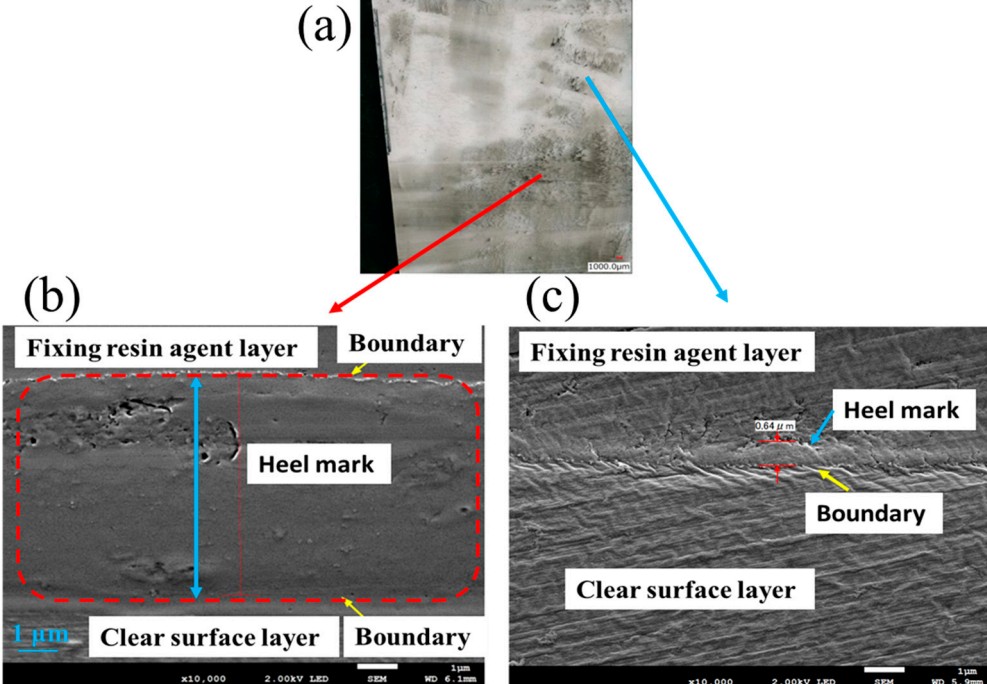

**Figure 5. Cross-sectional floor tile SEM results.** (**a**) Photographs of the floor surface before and after HM removal. The bottom of the photo show shows the bonded HM, while the top area shows the effects of our surface treatment. (**b**) Cross-sectional photo of the HM on a floor tile (**c**) Cross-sectional photo of tile with the HM removed. The SEM images were taken at magnification of 10,000.

The stained areas of the sample tiles were obtained from image analyses of the photographs using ImageJ analysis software (NIH, Bethesda, MD, USA). The analyses were performed by counting pixels with shading values for the HM stains using an arbitrary unit. Image analyses were carried out before and after the plasma treatments, and the resulting values were identified as A and B, respectively.

Figure 3b shows an HM stain before plasma irradiation, while Figure 3c shows the same stain after plasma irradiation. The A and B change was also calculated, and each experiment value is defined as the rate of change, which is given by

$$Rate\ of\ change = \frac{B}{A} \quad for\ A \leqq B \tag{1}$$

### 2.4. Electrical Measurements

The voltage values at the electrode position were measured by using a P6015A high-voltage probe (Tektronix, Inc., Beaverton, OR, USA), while circuit currents were measured

using a Current Monitor Model 3972 (Pearson Electronics, Palo Alto, CA, USA). The current and voltage waveforms were recorded by using a DPO4054 oscilloscope (Tektronix) at a temporal resolution of 10 µs and a voltage resolution of 5 V.

The pulse frequency was set at 1 kHz, and it was determined that a wave packet with a duration of 0.6 ms consisted of 24-sinusoidal waves. A number of the spike discharge occurrences per wave packet was counted in the recorded waveforms.

### 2.5. Temperature Measurements

The floor sample temperatures were measured every 60 s during the treatments via an MT-9 infrared thermometer (Mother tool) under dark conditions and 47% humidity.

## 3. Results

### 3.1. Stain Removal from Floor Samples

In this section, the results of using plasma discharge combined with pressured air to treat sample floor surfaces are analyzed in detail. Figure 3a–c shows photographs of (a) the initial (control) floor sample and (b) the HM taken before the experiment Condition C experiment, in which the two electrodes were separated by a 10 mm gap. Figure 3c shows a photograph taken of the HM during the plasma treatment. Here, the bright discharge arc between the electrodes can be clearly seen.

Figure 3d–f shows SEM images of the samples corresponding with Figure 3a–c taken at magnification of 2000, while Figure 3d shows the initial floor sample surface can be observed. In Figure 3e, which shows a close-up view of an HM before plasma treatment, it can be seen that the tile surface has been smoothened by polymeric coat deposits left by the shoe heel. However, in Figure 3f, which shows the same view after the treatment, polymer coat deposits are degraded. From these results, we can conclude that HMs polymeric deposits can be decomposed by the plasma treatments.

Table 2 shows a list of elemental components (in percentages) of the floor samples analyzed using the SEM energy dispersive X-ray (SEM-EDX) analysis of the same regions shown in Figure 3d–f. The rectangle area measured by SEM-EDX was approximately $40 \times 60$ µm$^2$. Note that the PVC base resin of the floor samples contains chlorine. Plasma irradiation reduces the C component of HM from 72% to 63%; however, bonding states are required by some analysis of XPS or FTIR. Figure 3e,f shows SEM image before and after the plasma treatment. No significant changes were observed. Surface roughness less than several hundred nanometers were still remained, however, the most content of the HMs were removed by the plasma treatments. The HM was seen dark area in Figure 3b. After the plasma treatment, it is clearly seen the HM-removed area turned to white color at location between the electrodes. This can be interpreted that the HM thickness was thinned by the plasma treatments. The floor component before and after the plasma treatment. Cl 6% arisen from the floor sample yielded 0% owing to coat of the HM. After removal of the HMs, chlorine concentration yielded 8% again, which indicates the HM could be removed by the 60 s plasma treatment.

**Table 2.** Floor surface composition. Elemental ratios in percentage analyzed by the SEM-EDX with a rectangle area of approximately $40 \times 60$ µm$^2$ with a magnification of 2000 are listed for control, HM, and removed HM. The chlorine originated from the PVC base polymer of the floor tile.

| Component | Control | HM | Removed HM |
|-----------|---------|-----|------------|
| C | 76 | 72 | 63 |
| O | 8 | 26 | 23 |
| Al | 10 | 2 | 1 |
| Cl | 6 | 0 | 8 |

Figure 4 shows photographs (a–d) and SEM images (e–p) of the floor samples before (a,e,i,m) and after the plasma-treatments for 60 s (b,f,j,n), 180 s (c,g,k,o), and 300 s (d,h,l,p). Even the long treatment time for 300 s, no damage on the floor sample was observed. Thus, the spark discharges with the distance of 10 mm and airflow with 20 slm are efficiently removed the HMs.

The cross-sections of the floor samples at locations with or without the HMs were observed by SEM at magnification of 10,000×, as shown in Figure 5b,c. Figure 5b shows a cross-sectional SEM image of the floor surface covered with HM polymeric deposits at a thickness of approximately 6 μm. Here, it should be noted that the observed HM thicknesses ranged from 5 to 15 μm and that HM surface deposits were fixed in place with a resin agent to suppress cross-sectional deformations during sample cutting. Figure 5c shows a SEM image of a cross-section of the plasma-treated floor surface where HM deposits with a thickness of less than 0.64 μm can be observed, thus indicating that almost the polymeric deposits had been removed from the floor surface by the applied atmospheric pressure plasma combination.

### 3.2. Improved Efficiency in Removing HM

Next, the efficiency of HM removal via atmospheric pressure plasma was tested under the conditions shown in Table 1. For simplification, three conditions: a 5 mm gap without airflow (A), a 10 mm gap without airflow (B), and a 10 mm gap with airflow (C), were compared in terms of HM removal efficiency. For all three conditions, the 20 SLM airflow was emitted from 35 mm above the sample surface, and the atmospheric pressure plasma irradiation period was fixed at 60 s.

For all of the tested conditions, the applied peak-to-peak voltages to the electrode were measured at 11.0, 20.0, 28.0, 33.2, 38.0, 40.0, 42.6, 44.2, 47.4, and 49.0 kVpp. Photographs were taken before and after the treatments, and removal efficiency was evaluated by image analyses of the HMs at the treated position. Brightness change rates were also summarized.

Figure 6 shows the brightness change dependence levels for applied voltages from 40 to 49 kVpp. For Condition A, it can be seen that the brightness changes decreased and increased with increasing applied voltage, reaching a maximum value of 1.11 at an applied voltage of 42.6 kVpp, and a minimum value at an applied voltage of 49 kVpp.

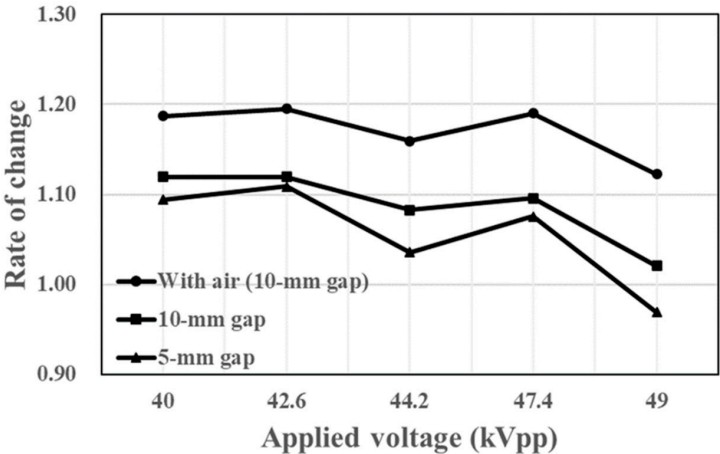

**Figure 6. Dependence of HM removal rates on applied voltages for three conditions.** The change rates were observed by analyzing brightness changes at the HM removal positions in photographs taken before and after the plasma treatment for 60 s.

For Condition B, where the electrode gap was widened from 5 to 10 mm and airflow was not supplied, brightness levels also decreased and increased with increasing applied voltage, reaching a maximum brightness value of 1.12 at an applied voltage of 42.6 kVpp, and showed a minimum value at an applied voltage of 49 kVpp. Hence, the 10 mm gap for Condition B showed brightness changes that were 0.01 higher than Condition A.

For Condition C, which had both applied airflow and a 10 mm gap, brightness levels decreased and increased with increasing applied voltage, reaching a maximum value of 1.19 at an applied voltage of 42.6 kVpp, and a minimum value at an applied voltage of 49 kVpp. Hence, we can see the combination of applied airflow and the 10 mm gap resulted in brightness changes that were 0.07 higher than those seen in Condition B.

These results show that the brightness level changes were highest when the airflow was applied to a 10 mm gap and that the changes among the three conditions were most significant at the applied voltage of 44.2 kVpp.

### 3.3. Electrical Waveforms

Since differences were noted in the discharge voltage waveforms (Figure 7), we conducted a detailed analysis. Figure 7a–c shows a typical discharge waveform at the applied voltage of 44.2 kVpp. The voltage waveforms for Condition A are shown in Figure 7a, where the solid black lines indicate the supplied voltage waveforms without connection to any load, and the solid red lines display the voltage waveforms with the electrodes connected for discharges. From the beginning of the test until 0.2 ms, we can see that the 40-kHz sinusoidal voltage increased and then suddenly dropped to below 1 kVpp. Thereafter, the voltage waveforms increased again.

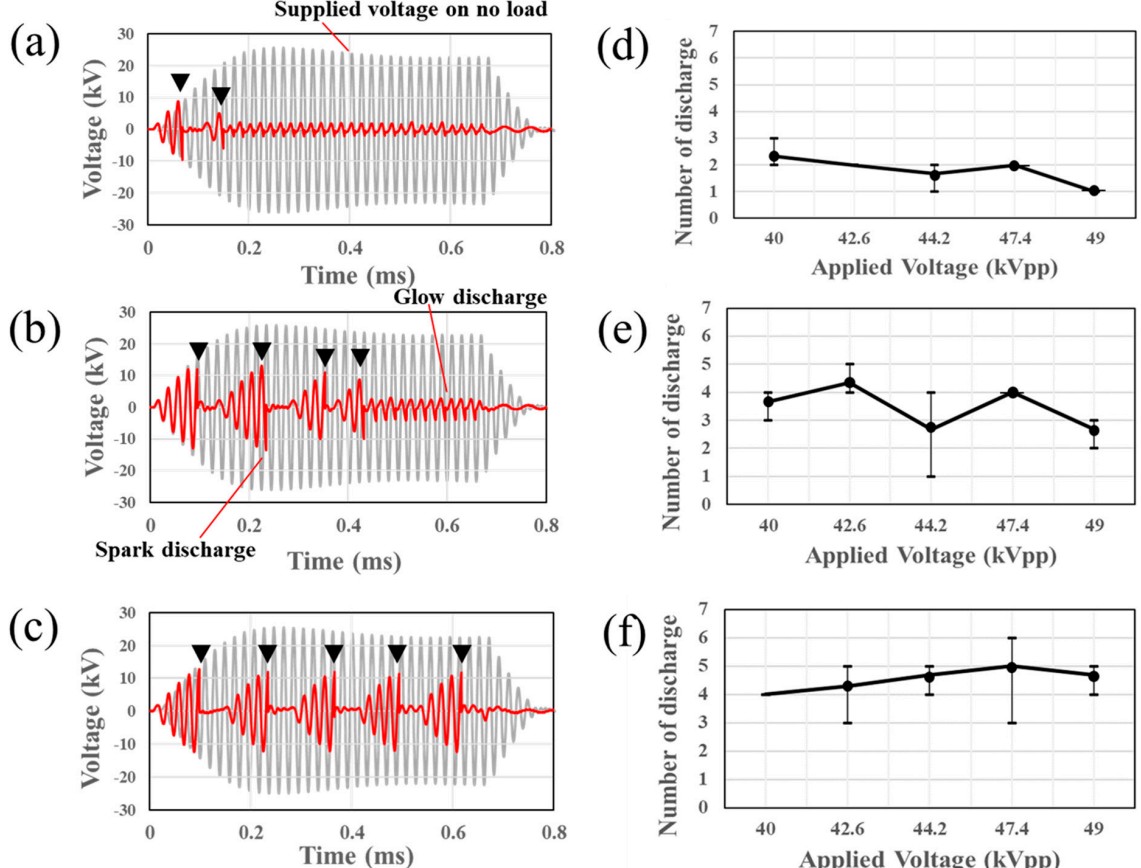

**Figure 7. Observed voltage waveform and discharge number.** The applied voltage waveform without a load connection was plotted in solid black lines, while voltage waveforms with electrode connections were plotted in solid red lines. (**a**) Voltage waveform at 5 mm gap without airflow. (**b**) Voltage waveform at 10 mm gap without airflow. (**c**) Voltage waveform at 10 mm gap with airflow. A voltage drop appeared on the waveforms, and current levels increased suddenly. This is called the spark discharge and is counted in 1 ms time units. (**d**) Graph showing number of discharges at 5 mm gap without airflow. (**e**) Graph showing number of discharges at 10 mm gap without airflow. (**f**) Graph showing number of discharges at 10 mm gap with airflow.

Since one case was a repeat of the sudden drop and the other was a constant voltage oscillation within a few kVpp, for simplicity, the former is called a spark discharge, and the latter is referred to as a glow discharge. Figure 7b shows a typical voltage waveform for Condition B. In this figure, four spark discharge groups could be counted, and a glow discharge was sustained after a period of 0.4 ms. The discharge voltage waveform for Condition C is shown in Figure 7c, where five spark discharge groups were observed, but no glow discharge was visible.

Figure 7d–f shows summaries of the spark discharge groups for Condition A (d), B (e), and C (f) at the applied voltage of 44.2 kVpp. From these results, we can see that the number of spark discharges increased in the order of Condition A (5 mm gap without airflow), B (10 mm gap without airflow), and C (10 mm gap with airflow).

Next, surface temperatures during plasma treatment were examined. Note that, as shown in Figure 8, the chemical decomposition of the HM staining was negligible because the floor surface temperatures were low (32.6 °C at maximum), which means there was no heat effect from the electric discharges.

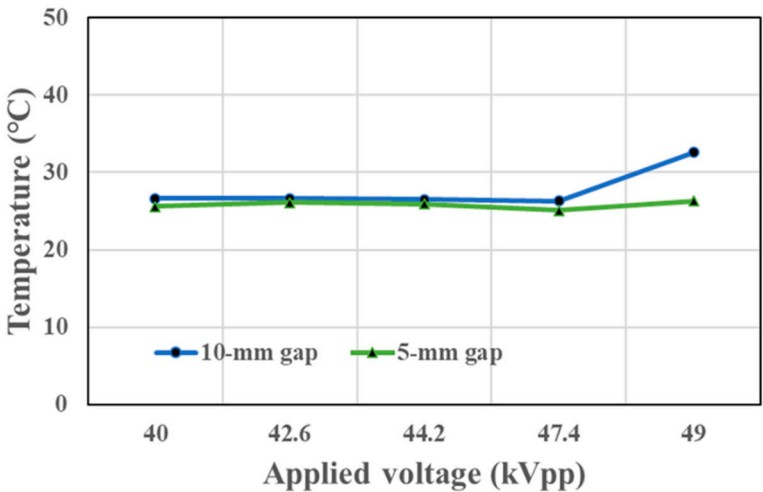

**Figure 8. Floor surface monitored by infrared thermometer.** The measured maximum temperatures are plotted based on the dependence of the applied voltages ranging from 40 kVpp to 49 kVpp. The conditions shown here were 5- and 10 mm electrode gaps without airflow.

## 4. Discussion

This study shows that HMs can be removed by spark discharges combined with pressurized air. Based on the removal rate results shown in Figure 6 and the number of discharges shown in Figure 7, the correlation between the removal rate and the number of discharges is summarized in Figure 9. Figure 9 also shows that the distance between the two electrodes and the presence or absence of pressurized air affected the HM removal rate. It was also found that increasing the gap from 5 to 10 mm increased the number of spark discharges, thereby increasing the HM removal rates, even without the presence of airflow (Condition B), but that adding airflow (Condition C) increased rates even further.

Compared with the plasma jet operation, the electric field of the spark discharge was 3 times larger at 221 V/m than that of the plasma jet at 75 V/m. Accelerations by the high electric field are essential factor for removal of organic components. Since the dielectric constant of the flooring material (PVC) of 4.55 is greater than the dielectric constant of air 1, creeping discharge and surface flashover occur likely on the floor surface, owing that charge accumulated locally on the floor by contacting the electrodes on the floor. When creeping discharge and tracking phenomenon occur on carbon-based dielectrics, pyrolysis and liberation of carbons occur with formation of conductive carbonization and degradation of electrical insulation [33,34]. In Figure 3, electrical currents parallel to the floor and perpendicular to the floor flowed through the HM. Therefore, an increase in

the number of spark discharges correlated with the removal rate of the HMs as shown in Figure 9.

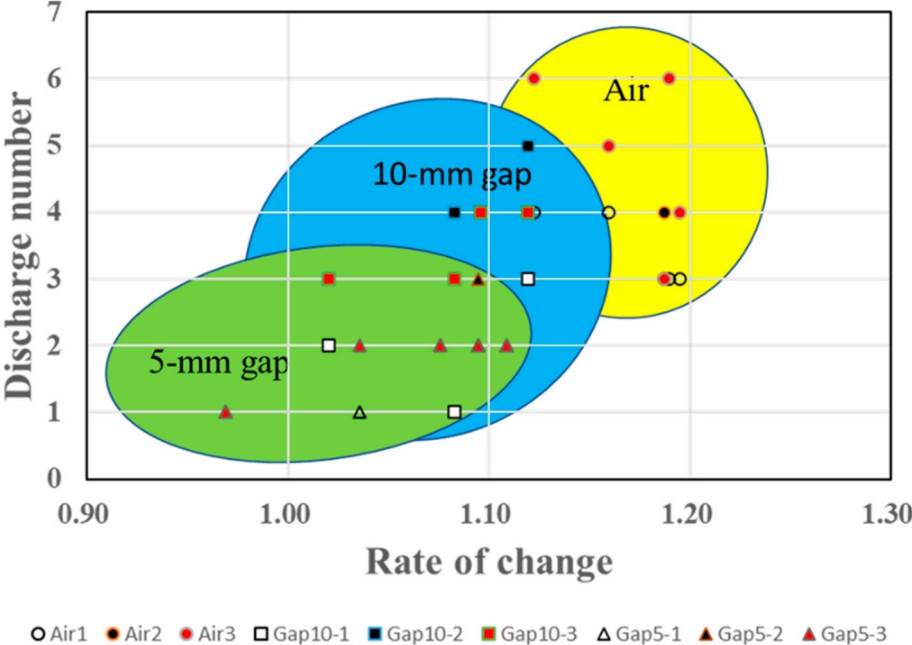

**Figure 9. Relationship between removal rate and the discharge number.** The removal rates of the HM depend on the number of spark discharges. The condition of airflow and 10 mm gap was the fastest removal of the HMs.

When the electrode gaps were widened from 5 to 10 mm, the weak electric field decreased electron acceleration, and ionization multiplication decreased electron density levels, thus making discharge ignitions difficult. Additionally, glow discharges were rarely observed unless the gap was narrowed. A similar phenomenon was reported at the ambient temperature, and shifting from a spark discharge to a stable glow discharge was reported to be difficult as well [35]. At an electrode gap of 10 mm, it was also found that the number of spark discharges was higher in the presence of pressurized air than without.

We currently conjecture that once a discharge has occurred, freed electrons remain localized and facilitate ionization. As a result, the reduced temperature between the electrodes produced by the airflow retards ignition discharges that can lower electron density. Therefore, glow discharges are suppressed, and the number of spark discharges increases. In order to clarify this hypothesis, we will perform measurements of electron density and species in the plasmas in our future studies.

For the purpose of floor cleaning, it is enough large 10 mm. Additionally, a gap distance of 10 mm is the limit determined from the maximum output of 49 kVpp from the power supply used in the experiment, in relation with ignition of a discharge. To widen the gap distance is needed to increase the applied voltage more than 49 kVpp. In our perspectives, a robot of vacuum cleaners recognizes automatically HM by cameras, and this spark can remove the spatially located HM with a size around 10 mm. For an examples, an floor can be cleaned by running a robot vacuum cleaner for 24 h a day.

Finally, since HM removal effectiveness is roughly in proportion to spark discharge amounts, we can also conjecture that increasing the number of spark discharges would contribute significantly to faster HM stain removal from floor surfaces.

## 5. Conclusions

Herein, we showed how bonded HM floor stains could be removed by spark discharges combined with pressurized air emitted between two electrodes that were placed in contact with the floor surface. We also showed that the HM removal rates increased

when the electrode gap was widened from 5 to 10 mm. Then, by comparing conditions with and without airflow, we showed that the application of pressurized air significantly increased HM removal rates. This is hypothetically attributed to the increase in the number of spark discharges.

From these results, we concluded that placing the electrodes 10 mm apart in contact with the floor surface while simultaneously emitting pressurized air between them at a flow rate of 20 SLM showed the best HM removal rates observed in this study primarily due to an increase in a number of spark discharges.

**Author Contributions:** Conceptualization, M.H., H.H., K.I., T.T. and Y.S.; methodology, M.H., K.I., T.T. and Y.S.; validation, M.H., K.I., T.T. and Y.S.; formal analysis, M.H., K.I., T.T. and Y.S.; investigation, M.H., H.T., K.I., T.T. and Y.S.; resources, M.H., H.T., K.I., H.H., T.T. and Y.S.; data curation, M.H., H.T., K.I., T.T. and Y.S.; writing—original draft preparation, M.H., K.I., T.T. and Y.S.; writing—review and editing, M.H., H.T., K.I., T.T. and Y.S.; visualization, M.H., H.T., K.I., T.T. and Y.S.; supervision, M.H. and H.T.; project administration, M.H. and H.T.; funding acquisition, M.H. and H.T. All authors have read and agreed to the published version of the manuscript.

**Funding:** A Grant-in-Aid for Specially Promoted Research (No. 19H05462).

**Institutional Review Board Statement:** Not applicable.

**Informed Consent Statement:** Not applicable.

**Data Availability Statement:** Not applicable.

**Acknowledgments:** The authors would like to thank Hiroki Kondo, Makoto Sekine, and Anna Komaki for their fruitful discussions.

**Conflicts of Interest:** The authors declare no competing financial interest.

## Abbreviations

**HM**, Heel marks; **PVC**, Polyvinyl chloride.

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
