# Peer review of "Indoor Floor Heel Mark Removal Using Spark Discharges and Pressurized Airflow"

_coatings, doi:10.3390/coatings12121938_

Round 1
Reviewer 1 Report
Due to the high quality of the papers published in the Coatings journal I have some major concerns regarding this manuscript because there are several drawbacks.
The authors should consider answering the following questions to avoid this issue and to find the “generalization” traits of their study: What new findings provide their study compared to other similar studies performed? What lessons could be learned from their study? What is the rationale of the study, limitations, and future applications of the results? The contribution of the paper is fuzzy. Please make a list that clearly states the actual contribution of the paper. It is not clearly presented, what is original in this paper.
The presented idea is quite interesting but unfortunately it is not well argued from a scientific point of view.
I appreciate the authors' results, but still, even if they say that these are preliminary results, in this form the work requires improvements before it is accepted for publication.
For example, new “traditional” cleaning methods prove to be quite effective and should be demonstrated if the floor tiles could be affected "following plasma treatment". Will the new scratches that will appear, be as easy to clean without affecting the flooring layer? From Figure 3 it is not clear how affected the floor tile is and whether it could be used at least in the same conditions as before the treatment. On line 59, it's not clear what it means “strong pressure levels”. Was the floor tile simply scratched under various pressures? Were they measured?
The introduction does not sufficiently motivate the originality and motivation of the study (even the experimental section should contain more details), so in the present form, this article is just a good case study.
Reviewer 2 Report
The work is devoted to an interesting question of the use of plasma. Domestic use for surface cleaning is an important area to introduce plasma techniques into everyday life for the benefit of society. This work is interesting in this regard. It does not open supernova opportunities in terms of fundamental work, but from the point of view of practice it is interesting. Contains new results. Well framed and understandable.
From the comments, it should be noted that the issue of scaling the methodology should be discussed in more depth, since spark methods are difficult to scale for practical general use (and the main value of the work lies in practical use, in my opinion).
Reviewer 3 Report
Dear author,
The paper "Indoor floor heel mark removal using spark discharges and pressurized airflow" is very good and logical. And also, it is significant to study how spark discharges combined with pressurized airflow in 60-s discharge treatments can remove heel marks. This paper has been reviewed but it needs minor revision before accepted. The followings are the points need to modify.
1. In abstract, it should have some results of the paper.
2. Figure 5~8, the labels of the figure should be the same color and font size of the paper.
3. In figure 6 d, e, f, the number of discharges should marked in the figure. Also, in figure 7.
4. In conclusion, if it needs further consideration of physics, I suggest finish the experiment and add the result. Otherwise, the paper is not finished. Or just delete the sentence.
5. Also, the author should cite the lately reference, most of the reference is 10 years ago. There should be more new paper about the topic.
Round 2
Reviewer 1 Report
The authors introduced figure 4 in the paper, at the end of the work, but in the text, it is not inserted. This revised form is sent in a different format than the initial one and the changes made by the authors are difficult to follow in this revised form. For example, in the revised form table 1 is not found.
I ask the authors to resubmit the paper with track changes on the initial format and to bring additional explanations if certain initial aspects do not want to be included in the revised version.
